# Ability to Glue Portuguese Eucalyptus Elements

**Aiuba Suleimana [1], Caroline S. Sena [2] , Jorge M. Branco [3],* and Aires Camões [4]**

[1]   Department of Rural Engineering, University of Lúrio, Unango-Niassa 1115-04, Mozambique;
    asuleimana4@gmail.com
[2]   Federal University of Bahia, Salvador 40170-110, Brazil; klorac@hotmail.com
[3]   Civil Engineering Department, Institute for Sustainability and Innovation in Structural Engineering,
    University of Minho, Campus de Azurém, 4800-058 Guimarães, Portugal
[4]   CTAC, Department of Civil Engineering, University of Minho, 4800-058 Guimarães, Portugal;
    aires@civil.uminho.pt
*   Correspondence: jbranco@civil.uminho.pt; Tel.: +351-253-510-200

**Abstract:** Portuguese forests have changed in recent years. These changes were mainly boosted by the wildfires that affected a significant percentage of the softwood area. Eucalyptus is actually the dominant wood species in Portuguese forests. This is not a native hardwood, but is being planted mainly for pulp and paper production, and its availability and mechanical performance have made it very present in timber construction in the last 50 years. Within the discussion to substitute imported raw materials, mainly from softwoods, with local hardwoods for the production of engineered wood products, the study of the ability to glue eucalyptus has become a necessity. This paper presents experimental works aimed to assess the ability to glue eucalyptus elements for the production of glued laminated timber (GLT) and cross-laminated timber (CLT). Since this wood species has been known for being difficult to dry, a preliminary study on the dimensional stability under moisture content variation was performed. Then, shear strength tests were made in accordance with ASTM D143. The objective was to correlate those results with the tests performed in the following research step. In this further stage, shear strength tests of the bond line were performed following EN 14080 and EN 16351. The results obtained in all the experiments show that eucalyptus has the potential to be glued and therefore the production of GLT and/or CLT using this local undervalued wood species is potentially of high industrial interest.

**Keywords:** eucalyptus; experimental evaluation; dimensional stability; shear strength; bond line

## 1. Introduction

The increasing political and social demands regarding the use of eco-friendly building materials are leading in recent years to a strong rise in the use of wood due to its $CO_2$ absorption capacity and the quality of renewable natural resource [1]. In recent studies of the UK construction sector, it has been shown that novel off-site panelized modular timber frame systems can save up to 50% of embodied carbon and 35% embodied energy when compared with traditional residential building methods and materials [2]. In addition, new material developments and, in particular, the efforts put into the prefabrication of all construction processes [3], ensuring a better quality, are the key drivers of timber construction [4].

Timber for construction is one of the many forest products used around the world. In fact, construction-grade timber and engineered forest products are some of the highest value products from trees [2]. There is growing interest in Europe towards glued laminated structural products made of hardwoods due to several reasons, such as the shortage of softwoods, large stocks of hardwoods and policies of re-afforestation for several hardwood species due to better adaptation to soil and

climate conditions. In most cases, these products reach greater bending strengths than those of the highest European softwood glulam strength classes [5], usually made of spruce or pine (GL24 to GL32). Moreover, hybrid and composite systems, as beam elements of sandwich panels, represent competitive solutions due to their improved durability, high strength-to-weight ratio and cost-effectiveness [6].

Portuguese forests, boosted by the wildfires that affected a significant percentage of the softwoods area, have changed and now are mainly composed of hardwoods which represented 69% of the forest area in 2010, eucalyptus (*Eucalyptus globulus Labill*) being the most abundant species (26%) [7]. Eucalyptus is a not native hardwood presenting fast growth, but has been planted mainly for pulp and paper production, as its availability and mechanical performance has made it very present in timber construction in the last 50 years. The potential of its use in structural applications is largely unknown, despite the evidence proved by the built heritage in the last 50 years, in particular, in Northern Portugal and Galiza. In fact, the recognized difficulties of this hardwood to be sawn and dried are important limitations to its use in structural applications [8].

The most significant efforts to promote the use of eucalyptus in load-bearing structures comes from Spain, in particular, from Galiza. For example, Touza Vásquez and Saavedra [9] presented a proposal for drying eucalyptus. The proposal consisted in having a drying process, divided into two phases, adopting a slow drying process to ensure a high material quality with a reduced presence of the expected cross-section collapse (drying fissures). Alvite et al. [10] have investigated the physical and mechanical properties of eucalyptus grown in Spain. The study addressed both sawn wood and glued laminated timber. Average values of 760 kg/m$^3$, 20,580 MPa and 130 MPa were presented, respectively for density, modulus of elasticity in bending and bending strength for sawn wood, while for glued laminated timber, the average values obtained were 20,300 MPa and 125 MPa, for modulus of elasticity in bending and bending strength, respectively.

Lopez-Suevos and Richter [11] have developed a study of new applications of eucalyptus. The study focused on glued laminated timber and in the use of primers to enhance the bond durability of eucalyptus. In this study, delamination and shear strength tests have been performed, with a significant improvement in the bonding quality being observed when polyurethane reactive (PUR) adhesives are used.

In 2011, ref. [12] conducted a study to optimize the manufacturing process, from the sawing, drying, cutting and finishing of eucalyptus grown in Portuguese forests. In terms of mechanical properties, ref. [13] revealed lower mechanical properties for sawn wood (75 × 75 mm$^2$): average values of 18,150 MPa for modulus of elasticity in bending and 75 MPa for bending strength. Before that, the National Laboratory of Civil Engineering (LNEC) conducted a large experimental program using clear wood specimens [14], with average values of 127 MPa for the bending strength and 17,500 MPa for the modulus of elasticity in bending being reported.

Recently, Martins et al. [15] performed research with the main goal to assess the potential of eucalyptus timber for load-bearing structures. Glulam beams made of eucalyptus and hybrid ones, mixing eucalyptus with poplar have been evaluated. The obtained bending strength and modulus of elasticity are above the typical values found in the literature for the most common hardwoods available in European forests [15].

The Portuguese eucalyptus shows potential to be used in engineered wood products, like glulam and cross-laminated timber (CLT). It is an available hardwood, with a reasonable cost. Taking into account its mechanical properties, classified as D40 by the EN 1912:2012 [16], the improved engineered wood products that can be produced will for sure contribute to add value to Portuguese forests. However, a wide and detailed research is needed to improve the characterization of the raw material (eucalyptus lumber) and to assess the bond line quality. Moreover, the known difficulties to dry this hardwood must be solved. This paper presents a first step in a wide research program to assess and develop the potential use of local hardwoods species in structural applications. The focus is engineered wood products, namely glued laminated timber (GLT) and cross-laminated timber (CLT), as they represent added value products that can have a direct effect from the national industry perspective. In

particular, the modular construction of building structures has attracted significant attention from the construction industry because of their many advantages over traditional construction methods, it being demonstrated that timber-framed modules are suitable for medium-rise buildings [17]. The study started with the assessment of the dimensional stability of small samples in accordance with NP 615 [18]. Eucalyptus is a wood species known for being very difficult to dry and its dimensional stability can somehow jeopardize glued elements. Therefore, it is essential to assess the dimensional stability of eucalyptus when submitted to variations in moisture content because it can compromise the gluing process and the final medium- and long-term behavior of the glued elements. Then, a series of shear tests were performed. The bond line quality was evaluated through shear tests following the EN 14080 [19] adopting surface-bonding specimens (GLT) and edge-bonding ones (CLT) following the EN 16351 [20]. The aim of those tests is to assess the ability of eucalyptus to be glued using an industrial process in the production of glued elements such as GLT and CLT. Moreover, and because this mechanical property can be directly correlated with the shear strength of the bond line, shear strength tests on eucalyptus specimens were made according to ASTM D143 [21].

## 2. Experimental Work

For each series evaluated in this experimental program, all specimens were kept in a climatic chamber under controlled environmental conditions of 20 °C and 65% relative humidity (RH) for approximately 2 weeks until their stabilization was reached. The stabilization process was considered complete when the difference between two successive weight measurements, within a space of 6 h, was less than 0.5% [22]. Tests were performed only after this stabilization process. After each test, a small sample near the rupture zone was collected and used to quantify the density (EN 384 [23]), and the moisture content (EN 13183-1 [24]) of each specimen was also tested. In the further sections, more details about each experimental program are presented and discussed.

### 2.1. Dimensional Stability

For the determination of the dimensional variation, the procedure described by ISO 13061-1 [25], using Equation (1) for the determination of shrinkage ($\varepsilon$), was followed:

$$\varepsilon = \frac{l_1 - l_2}{l_2} \times 100 \tag{1}$$

where $l_1$ is the size of a given direction of a saturated specimen and $l_2$ is the size in the same direction for the specimen in anhydrous state.

Square specimens with $50 \times 50 \times 10$ mm$^3$ dimensions were used to measure the linear shrinkage in the radial and tangential direction, as shown in Figure 1. Due to the thickness of the samples, the longitudinal direction is neglected as expected. During the preparation of the specimens, special attention was paid to ensure an adequate definition of the radial and tangential direction of the annual rings, as can be depicted in Figure 1.

After the stabilization process already described, each of the specimens was weighed and measured using the test setup developed for this particular purpose (see Figure 2).

Then, specimens were immersed in water for 2 weeks ensuring that they were completely saturated after this period. After, the specimens were weighed and measured again before being kept in the climatic chamber under controlled environmental conditions (20 °C and 65% RH) until they reached stabilization (as defined by ISO 3130 [22]). This cycle was repeated one more time, after which, specimens were dry in an oven at a temperature of 103 °C until stabilization was reached. Then, final weight and measurements were taken to quantify the dried mass and volume of each specimen. Using the mass and volume when stabilization (dry specimens with constant weight) was reached, it was possible to quantify the density ($\varrho$) of each specimen.

Table 1 summarizes the main results obtained in the tests aimed to evaluate the dimensional stability of eucalyptus samples. The reading values result from the average of the two cycles of

measurements made. First, it is important to point out that the density values of the specimens are homogeneous, with a coefficient of variation (CoV) of less than 2%, ensuring that they can be assumed as belonging to the same sample. On the other hand, also the values registered for the linear shrinkage do not show a significant variation, the CoV being around 12% and 14%. Taking into account the small dimensions of the specimens, the accuracy needed for the measuring system and the influence that annual rings' orientation can have on the results, the variation observed can be assumed as normal.

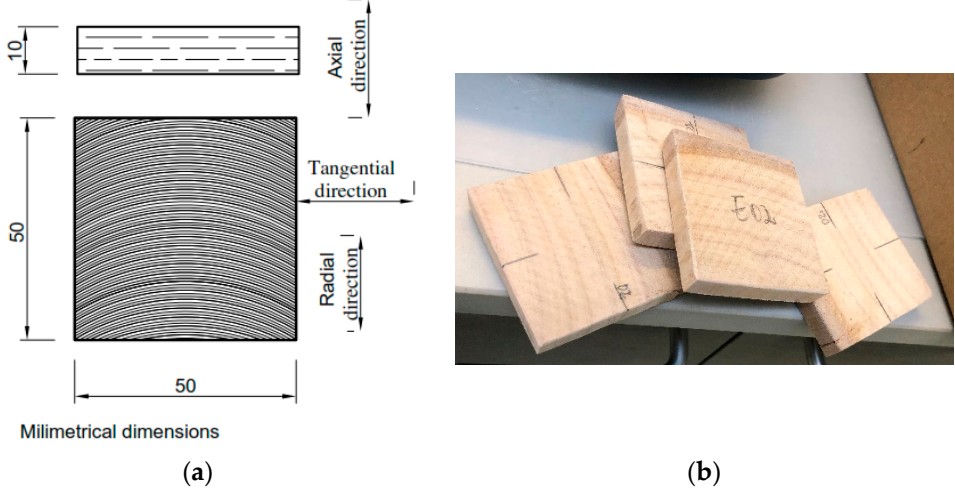

**Figure 1.** Specimens used for the dimensional stability evaluation. (**a**) Scheme and (**b**) photo.

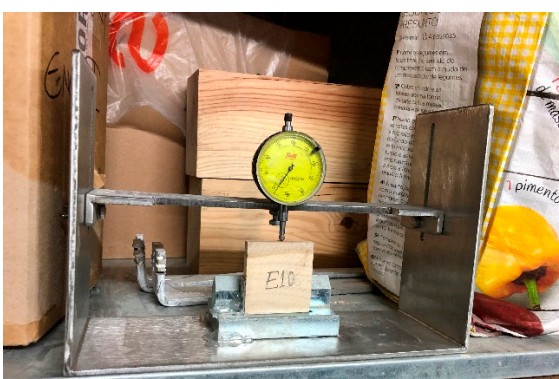

**Figure 2.** Test setup used to measure the specimens.

**Table 1.** Radial ($\varepsilon_R$) and tangential ($\varepsilon_T$) linear shrinkage measured on the eucalyptus wood specimens according to ISO 13061-1 [24].

| Specimen | Shrinkage (%) | | Density (kg/m³) |
| --- | --- | --- | --- |
| | Radial ($\varepsilon_R$) | Radial ($\varepsilon_T$) | |
| D.01 | 7.66 | 16.33 | 821.15 |
| D.02 | 7.99 | 16.60 | 838.39 |
| D.03 | 8.25 | 17.33 | 814.75 |
| D.04 | 8.61 | 14.59 | 822.17 |
| D.05 | 7.48 | 11.85 | 825.25 |
| D.06 | 8.31 | 19.48 | 808.23 |
| D.07 | 8.51 | 18.89 | 827.35 |
| D.08 | 10.71 | 16.24 | 801.40 |
| D.09 | 10.44 | 19.91 | 783.88 |
| D.10 | 8.24 | 18.39 | 821.15 |
| Mean | 8.62 | 16.96 | 816.37 |
| CoV (%) | 11.99 | 13.67 | 1.78 |

### 2.2. Wood Shear Strength

The shear strength of the eucalyptus wood was quantified since this mechanical property is important when evaluating the mechanical behavior of glued elements. Moreover, the bond line quality is dependent on the wood shear strength as all gluing processes aim to ensure that the failure happens on the wood side and not on the glue line. The test scheme and the geometry of the pieces followed the guidelines of the ASTM D143 standard [21].

A load cell with a maximum loading capacity of 200 kN was used and the experiments were realized with a loading rate of 0.01 mm/s, calibrated based on previous research. In total, 10 specimens were analyzed with the geometry outlined in Figure 3.

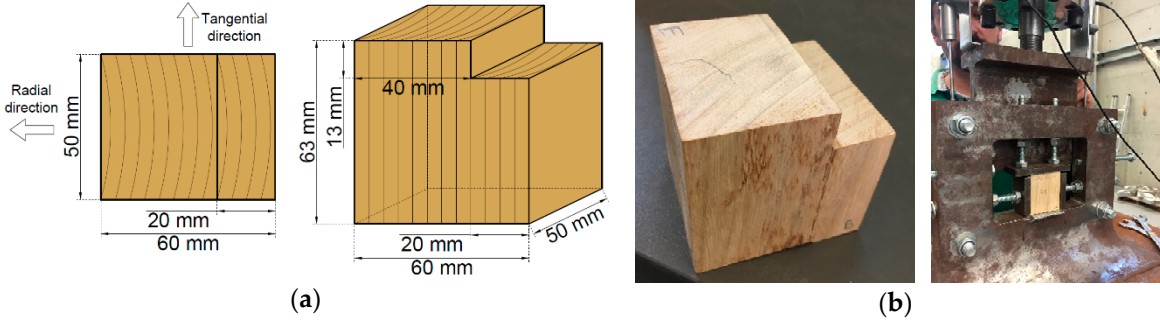

**Figure 3.** Geometry and test setup for the evaluation of the wood shear strength. (**a**) Geometry and dimensions in mm, (**b**) test sample.

The results obtained for the wood shear strength are presented in Table 2, where $F_{max}$ represents the maximum applied load and $f_v$ the respective shear strength. The shear strength was obtained considering the exact shear area that was measured for each specimen. In each test, the density ($\varrho$) was corrected for the reference value of the moisture content (12%), in accordance with EN 384:2016 [23].

**Table 2.** Tests results obtained for the eucalyptus shear strength ($f_v$).

| Specimen | $F_{max}$ (kN) | $f_v$ (MPa) | $\varrho$ (kg/m$^3$) |
|:---:|:---:|:---:|:---:|
| S01 | 25.52 | 10.21 | 864.73 |
| S02 | 31.17 | 11.99 | 799.15 |
| S03 | 31.03 | 11.93 | 852.50 |
| S04 | 31.39 | 12.56 | 880.27 |
| S05 | 30.46 | 11.95 | 845.87 |
| S06 | 28.27 | 11.31 | 913.31 |
| S07 | 33.00 | 13.20 | 863.82 |
| S08 | 33.12 | 12.99 | 855.43 |
| S09 | 28.95 | 11.35 | 873.10 |
| S10 | 27.78 | 10.68 | 851.60 |
| S11 | 31.73 | 12.20 | 834.89 |
| S12 | 28.73 | 11.27 | 785.74 |
| S13 | 32.72 | 13.09 | 859.62 |
| S14 | 37.33 | 14.64 | 922.22 |
| S15 | 31.48 | 11.88 | 834.60 |
| S16 | 31.60 | 12.64 | 891.61 |
| S17 | 30.31 | 12.12 | 848.34 |
| S18 | 34.06 | 13.36 | 871.06 |
| S19 | 28.21 | 10.85 | 850.86 |
| S20 | 32.18 | 12.62 | 849.74 |
| Mean | 30.95 | 12.14 | 857.42 |
| CoV (%) | 8.21 | 8.42 | 3.64 |

The tests results obtained show homogeneous shear stress values with a CoV below 9%. In addition, the density values present a small variation (CoV less than 4%), helping to consider that the specimens belong to the same sample.

### 2.3. Bond Line Shear Strength

The bond line connection between two eucalyptus elements was analyzed through two different experiments. One was with surface-bonding specimens replicating GLT structural elements and the other adopted edge-bonding specimens simulating what can happen in CLT manufacturing. Both experiments adopted the physical concept of applying shear stress on the glue line. For the surface bonding elements, Annex D of EN 14080 [19] was followed, submitting the glued connection in parallel to the grain shear test. In the case of the edge-bonding elements, the glued connection was evaluated under a shear test in the perpendicular grain direction, in accordance with Annex D of EN 16351 [20]. It is important to point out that the production of those two kinds of glued elements for timber structural applications, GLT and CLT, are normalized by EN 14080 [19] and EN 16351 [20], respectively, and the bond line shear test is one of the available methods suggested by those standards to evaluate the quality of the gluing process. Therefore, the bond line shear strength tests performed are a first step to prove the ability to use Portuguese eucalyptus in the production of structural glued elements.

Since in a further research step, glulam beams will be produced with eucalyptus lamellae of 22 mm, the specimens that were prepared adopted this thickness for the elements. Moreover, and to ensure the representativeness of the bond line evaluation, the specimens were prepared gluing the elements in the same conditions of the future glulam beams (hydraulic press, glue line thickness, level and time of pressure, curing, etc.).

Surface bonding specimens measured $50 \times 50 \times 178$ mm$^3$, gluing 8 elements of $50 \times 50 \times 22$ mm$^3$ and therefore ensuring 7 bond lines to test. Edge-bonding elements were composed of 5 elements of $70 \times 50 \times 22$ mm$^3$, glued together in specimens of $350 \times 50 \times 22$ mm$^3$ with 4 glue lines to be evaluated. It is important to note that not all the bond lines were evaluated because in some cases, the failure on the wood side compromised the following glue line. Figure 4 presents the geometry of the specimens used to evaluate the bond line quality.

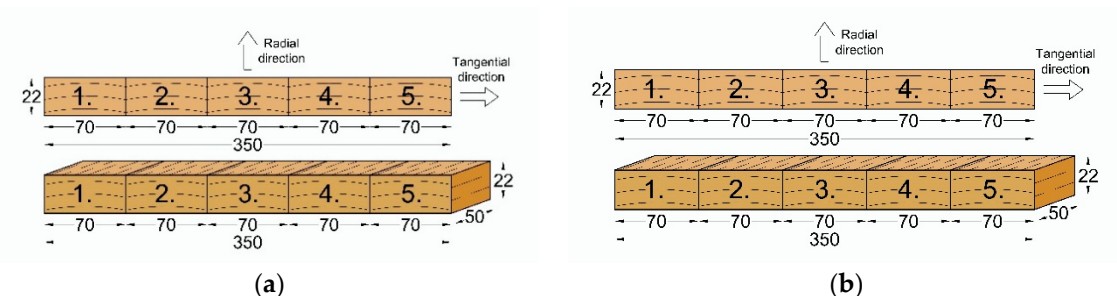

**(a)**　　　　　　　　　　　　　　　　　　　**(b)**

**Figure 4.** Geometry of the specimens used to assess the quality of the bond line. (**a**) Surface bonding and (**b**) edge bonding (dimensions in mm).

As glue, a melamine urea formaldehyde (MUF) resin system, namely the MUF 1247/2526 from AkzoNobel, was used. As already mentioned, the specimens were prepared in an industrial environment, by Rusticasa industry, adopting the current methodologies and procedures for the production of glued timber elements. After their production, the specimens were transported to the laboratory and kept in a climatic chamber under controlled conditions (T = 20 °C and 65% RH) until their stabilization in accordance with ISO 3130 [22]. Concluding the stabilization processes, the shear tests were performed adopting the setup defined by ASTM D143 [21] (Figure 5).

In each test, the maximum applied load ($F_{max}$) was measured to calculate the corresponding shear strength ($f_v$). The density ($\varrho$) and the moisture content (w) of the specimen at the test time were

registered. Figure 6 presents the experimental load–displacement curves obtained in the bond line shear tests performed, highlighting the mean curves (in black). The development of the force–displacement curves obtained express the brittle failure of the shear behavior of the glued connections.

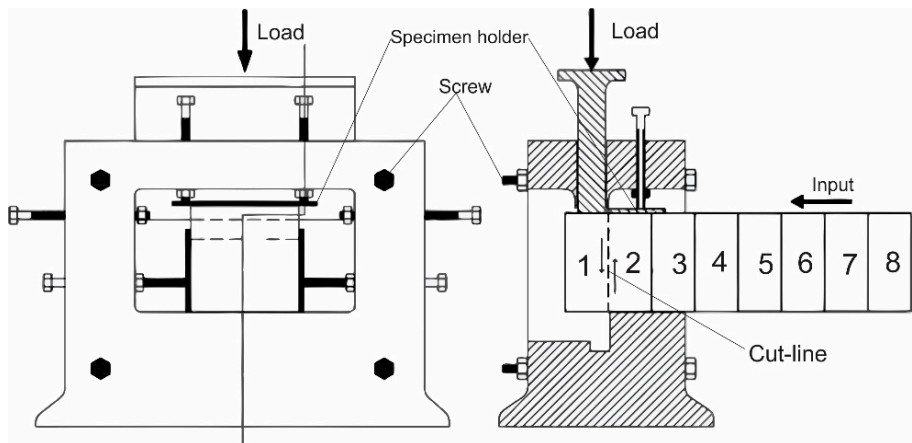

**Figure 5.** Test setup used to evaluate the bond line quality through shear strength tests.

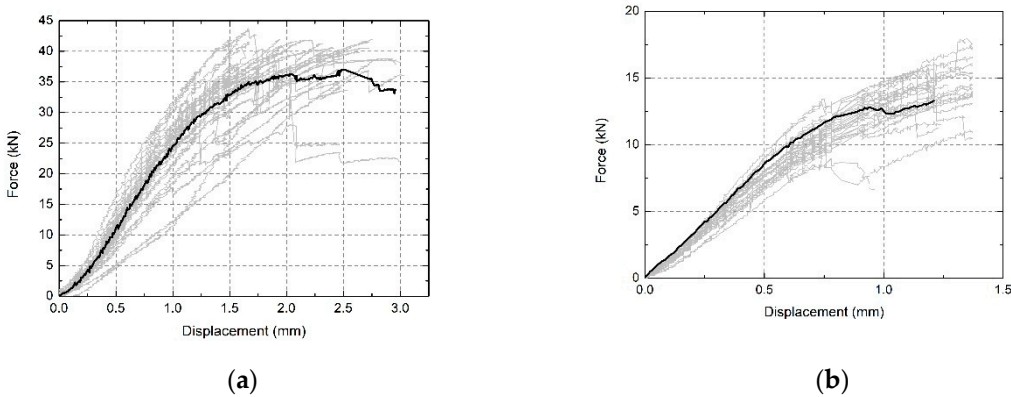

**Figure 6.** Load–displacement curves of the bond line shear tests, (**a**) surface bonding (Bi) and (**b**) edge bonding (Ei).

Tables 3 and 4 present the average results obtained in the shear tests for the surface- (Bi) and edge (Ei)-bonding specimens, using 28 and 30 specimens, respectively.

**Table 3.** Shear tests results obtained for the bond line evaluation using surface-bonding specimens (Bi).

| Specimens (Bi) | $F_{max}$ (kN) | $f_v$ (MPa) | $\varrho$ (kg/m$^3$) | W (%) |
|---|---|---|---|---|
| Mean | 36.80 | 14.19 | 738.87 | 11.64 |
| CoV (%) | 11.99 | 19.57 | 1.06 | 5.98 |

**Table 4.** Shear tests results obtained for the bond line evaluation using edge-bonding specimens (Ei).

| Specimens (Bi) | $F_{max}$ (kN) | $f_v$ (MPa) | $\varrho$ (kg/m$^3$) | W (%) |
|---|---|---|---|---|
| Mean | 12.32 | 12.03 | 712.72 | 11.41 |
| CoV (%) | 12.71 | 13.30 | 2.05 | 3.89 |

The tests results obtained show homogeneous values. Density values have a CoV no greater than 2.1%, contributing to the representativeness of the results obtained. In the same line, the moisture content of the specimens was around the reference value of 12% with a small CoV (5.98% and 3.89%).

Important to point out that all failures observed were on the wood side, while the glue line was always intact. Figure 7 depicts the typical failure modes on the wood side observed in both kinds of experiments realized.

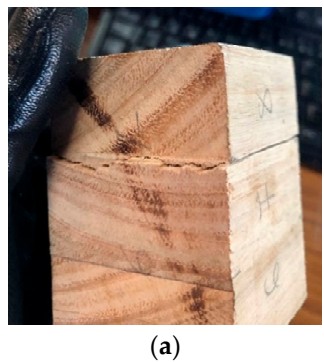

(**a**)

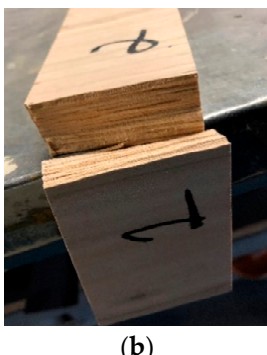

(**b**)

**Figure 7.** Typical wood shear failure observed during the shear tests performed. (**a**) Surface-bonding specimens (Bi) and (**b**) edge-bonding specimens (Ei).

## 3. Discussion

Here, a discussion is presented based on the experimental results obtained in the tests performed and compared with existing ones, when available. The results obtained for the linear values of the radial ($\varepsilon_R$) and tangential ($\varepsilon_T$) shrinkage are quite consistent with CoV values below 14%. The results obtained were 8.62% and 16.96%, respectively, being consistent with the rule of thumb that the tangential shrinkage is double the radial one. Another important fact to point out is the homogeneity demonstrated by the density of the samples (CoV smaller than 2%). Comparing with the literature, Alvite et al. [10] reported values similar to the ones here obtained, $\varepsilon_R = 7.5\%$ and $\varepsilon_T = 13.8\%$, but with a higher CoV (13 to 30% and 8 to 27%, respectively). This researcher reported density values of 760 kg/m$^3$ with a CoV of 11 to 20%, while the specimens used in this experimental work presented a density of 816.17 kg/m$^3$ and a CoV of 1.78%. As expected, the dimensional instability revealed by the eucalyptus wood, with values for the linear shrinkage higher than most of the softwoods (less than 4% [26]), can be a limitation to the use of this hardwood in the production of glued elements.

Analyzing the shear strength behavior, the value obtained in the experimental program, 12.14 MPa, is consistent with the literature: 16.2 MPa by [1], 15.4 MPa by [8] and 13.7 MPa by [14]. Interesting is the homogeneity exhibited by the tests results, i.e., a CoV of 8.42%, significantly smaller than the CoV equal of 19% presented by [1]. It is important to note that the value obtained for the shear strength is significantly higher than the characteristic value reported for the strength class D40 by EN 338 [27] (3.8 MPa). This higher value of shear strength, even when assuming the underestimated value reported by this standard for the D40 class, represents a great opportunity for eucalyptus to be used in glued elements. For example, an improved shear strength will result in a higher rolling shear resistance, the most important failure mode associated with CLT panels under bending. The same improvement, higher shear strength, represents an advantage of the glued-laminated timber elements made of eucalyptus.

Analyzing the results obtained in the shear tests on glued elements, the results are reliable. In all tests, the failure happened on the wood side, and the shear strength results for the glued elements (14.19 and 12.03 MPa) are consistent with the values obtained for the wood shear strength (12.14 MPa). This is particularly true, when taking into account that the density values of the glued samples are slightly smaller (738.87 and 712.72 kg/m$^3$) than the density of the sample used to evaluate the wood shear strength (857.42 kg/m$^3$). The literature lacks tests results on glued eucalyptus elements, but [11] reports values of 13.8 ± 1.4 MPa for the shear strength of glued elements of eucalyptus. Therefore, it can be assumed that the values obtained are aligned with the known values. In 2018, ref. [28] reported that eucalyptus CLT specimens bonded with a PUR adhesive had a maximum of 3.51 MPa. In addition,

these authors concluded that eucalyptus CLT bonded with a PUR adhesive has better mechanical bond performance.

## 4. Conclusions

This paper presents a first step in a wide research program to assess and develop the potential use of local hardwoods species in structural applications. The focus is the use of eucalyptus for engineered wood products, namely glued laminated timber (GLT) and cross-laminated timber (CLT), as they represent added value products that can have a direct effect from the national industry perspective. In this context, the ability to glue eucalyptus elements was assessed experimentally.

A preliminary study on the dimensional stability under moisture content variation was performed. Then, shear strength tests were made in accordance with ASTM D143. In this further stage, shear strength tests of the bond line were performed following EN 14080 and EN 16351. In conclusion, it is possible to point out the reliability reported by all test results. The higher dimensional instability supported by the linear shrinkage coefficients was expected. Eucalyptus is known for being a wood species difficult to work with, also having drying problems [10]. In terms of wood shear strength, the tests results are higher than the values suggested by EN 338 [27] for the D40 strength class. Further, all glued elements failed on the wood side. Therefore, it can be concluded that the bond line was effective. The experimental values here obtained for eucalyptus demonstrate the potential to use this hardwood to produce glued elements. The next step of the research will comprise full-scale bending tests on glulam structural elements. The objective is to verify the bending strength of the glulam beams made of Portuguese eucalyptus. Past research, e.g., Martins et al. [15], has pointed out mechanical properties (MoE of 23700 MPa and a bending strength of 121 MPa) above the values reported by the literature for spruce or pine (GL24 to GL32). However, additional research will be needed to sustain an accurate development of glued elements using this local hardwood. For example, the drying process of the raw material must be improved. Moreover, durability assessment and creep bending tests are needed. It is essential to ensure an adequate performance over the mid- and long-term.

For certain, the production of engineered wood products using local and underutilized hardwoods will promote the value and the preservation of Portuguese forests.

**Author Contributions:** Conceptualization and methodology, J.M.B. and A.C.; experimental work and data analysis, A.S.; writing—original draft preparation, A.S. and C.S.S.; writing—review and editing, J.M.B. and A.C. All authors have read and agreed to the published version of the manuscript.

**Funding:** This research received no external funding.

**Acknowledgments:** The authors would like to thank the Rusticasa company for gluing the wood elements.

**Conflicts of Interest:** The authors declare no conflict of interest.

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
