# Peer review of "Ability to Glue Portuguese Eucalyptus Elements"

_buildings, doi:10.3390/buildings10070133_

Round 1
Reviewer 1 Report
Dear authors,
this paper presents a very interesting starting point of a research. Eucalyptus may be used for constructions.
Nevertheless major revisions are needed.
First concern: this is configured more like a technical report than a scientific paper.
Introduction fails to properly discuss the paramount topic of timber buildings.As an example, two of the most cited and recent review on this topic are missing:
Caniato, et al. Acoustic of lightweight timber buildings: a review, Renew Sust
Energ Rev 80C (2017) pp. 585-596, DOI 10.1016/j.rser.2017.05.110
Ramage et al. The wood from the trees: the use of timber in construction. Renew Sust
Energ Rev 2017;68(Part 1):333–59
Please do include them (and others) and briefly discuss.
Experimental work is not properly introduced and and fails to explain WHY you are doing these tests: are they the necessary ones? If yes to do what? to which aim? if no, why did you choose only those ones?
In discussion you reported only at the end one sentence containing the comparison with spruce or pine. You say that "it is fine". How? what? where? every parameter? just one? please provide at least one comparing table with some more comments. Which tests do you think are needed more? thermal one? acoustic ones? moisture resistance ones? load ones? impact ones? please explain
Author Response
Point 1: Introduction fails to properly discuss the paramount topic of timber buildings. As an example, two of the most cited and recent review on this topic are missing: Caniato, et al. Acoustic of lightweight timber buildings: a review, Renew Sust Energ Rev 80C (2017) pp. 585-596, DOI 10.1016/j.rser.2017.05.110; Ramage et al. The wood from the trees: the use of timber in construction. Renew Sust Energ Rev 2017;68(Part 1):333–59. Please do include them (and others) and briefly discuss.
Response 1: The introduction was revised and an analysis on the topic of timber buildings was included. References were added, including the ones suggested by the reviewer.
Bellow it is possible to find examples of the text added to the introduction and the corresponding references.
In recent studies of the UK construction sector it has been shown that novel off site panelised modular timber frame systems can save up to 50% of embodied carbon and 35% embodied energy when compared to traditional residential building methods and materials [2]. In addition, new material developments and in particular, the efforts put in the prefabrication of all the construction process [3], ensuring a better quality, are the key drivers of the timber construction [4].
In fact, construction-grade timber and engineered forest products are some of the highest value products from trees [2].
In particular, the modular construction of building structures has attracted significant attention from the construction industry because of their many advantages over traditional construction methods, being demonstrated that timber-framed modules are suitable for medium-rise buildings [17].
- Ramage, M., Burridge, H., Busse-Wicher, M., Fereday, G., Reynolds, T., Shah, D., et al. (2017). The wood from the trees: the use of timber in construction. Renewable and Sustainable Energy Reviews, 68, pp. 333-359.
- Ferdous, W., Manalo, A, Aravinthan, T., Fam, A. (2018). Flexural and shear behaviour of layered sandwich beams. Construction and Building Materials, 173, pp. 429-442.
- Caniato, M., Bettarello, F., Ferluga, A., Marsich, L., Schmid, C., Fausti, P. (2017). Acoustic of lightweight timber buildings: a review. Renewable and Sustainable Energy Reviews, 80C, pp. 585-596, 10.1016/j.rser.2017.05.110
- Ferdous, W., Bai, Y., Ngo, T.D., Manalo, A., Mendis, P. (2019). New advancements, challenges and opportunities of multi-storey modular buildings – a state-of-the-art review. Engineering Structures, 183, pp. 883-893.
Point 2: Experimental work is not properly introduced and fails to explain WHY you are doing these tests: are they the necessary ones? If yes to do what? to which aim? if no, why did you choose only those ones?
Response 2: The explanation of the experimental work performed was complemented and all the tests done were justified with why are needed? and what it is the purpose?
Bellow some examples of the paragraphs added to justify the experimental work are listed.
So, it is essential to assess the dimensional stability of Eucalyptus when submitted to variations on the moisture content because it can compromise the gluing process and the final medium and long term behavior of the glued elements.
The aim of those tests is to assess the ability of the Eucalyptus to be glued using an industrial process in the production of glued elements such as GLT and CLT.
Moreover, the bond line quality is depending on the wood shear strength as all gluing process aims to ensure that the failure happens in the wood side and not in the glue line.
Point 3: In discussion you reported only at the end one sentence containing the comparison with spruce or pine. You say that "it is fine". How? what? where? every parameter? just one? please provide at least one comparing table with some more comments. Which tests do you think are needed more? thermal one? acoustic ones? moisture resistance ones? load ones? impact ones? please explain.
Response 3: The discussion part was updated providing a more detailed comparison of the glulam made of Eucalyptus and the current values showed by glulam made of Pine and Spruce. Moreover, the next steps of the research are presented. Bellow some examples of the paragraphs added to improve the discussion part.
The experimental values here obtained for the Eucalyptus demonstrate the potential to use this hardwood to produce glued elements. The next step of the research will comprise full-scale bending tests on glulam structural elements. The objective is to verify the bending strength of the glulam beams made of Portuguese Eucalyptus. Past researches, e.g. Martins et al. [15], have pointed out mechanical properties (MoE of 23700 MPa and a bending strength of 121 MPa) above the values reported by the literature for Spruce or Pine (GL24 to GL32). However, additional researches will be needed to sustain an accurate development of glued elements using this local hardwood. For example, the drying process of the raw material must be improved. Moreover, durability assessment and creep bending tests are needed. It is essential to ensure an adequate performance during mid and long terms.
For certain, the production of engineered wood products using local and underutilized hardwoods will promote the value and the preservation of Portuguese forests.
Reviewer 2 Report
Comments
This paper investigated glue-line shear behaviour of a Portuguese Eucalyptus elements. The outcome is interesting for readers. However, there are several aspects that need to be improved and explained. The reviewer can only recommend for publication if the author satisfactorily address the following comments in the revised version.
- The author need to present load-displacement curve of the bond-line shear test.
- Why this study selected bond line shear test for investigation?
- The author need to clarify the main variables of the study and what are their effect on the bond-line shear strength?
- The introduction section has not provided sufficient information on the glue laminated timber. Recently glue laminated timber was investigated for manufacturing railway sleepers [Ref: Flexural and shear behaviour of layered sandwich beams], modular buildings [Ref: New advancements, challenges and opportunities of multi-storey modular buildings–A state-of-the-art review] and bond performance [Ref: Bond behaviour of composite sandwich panel and epoxy polymer matrix: Taguchi design of experiments and theoretical predictions]. The author need to improve the Introduction section by citing the recently published research as referred in this comment.
Round 2
Reviewer 1 Report
Dear Authors, It is visible how you did improve the paper. Exspecially in the introduction section and in the materials and methods.
Nevertheless the discussion part is very poor and overlaps with the conclusions.
Still, I don't think it has a scientific form. Please split the discussion and conclusions in two two separate parts. Please do improve discussion because in this form it is just a technical report.
Author Response
The discussion was separated from the conclusions and has been improved.

Reviewer 2 Report
Thanks for addressing comments
Author Response

(The authors gave the same response as above.)

Round 3
Reviewer 1 Report
Dear Authors,
Thank you very much for your improvements. Still I think that discussions are poor but it is your paper... you decide how long and deep to discuss your results.